# Lizard Blastema Organoid Model Recapitulates Regenerated Tail Chondrogenesis

**DOI:** 10.3390/jdb10010012

**Published:** 2022-02-10

**Authors:** Ariel C. Vonk, Sarah C. Hasel-Kolossa, Gabriela A. Lopez, Megan L. Hudnall, Darian J. Gamble, Thomas P. Lozito

**Affiliations:** 1Department of Stem Cell Biology and Regenerative Medicine, Keck School of Medicine, University of Southern California, 1425 San Pablo St, Los Angeles, CA 90033, USA; vonk@usc.edu (A.C.V.); haselkol@usc.edu (S.C.H.-K.); galopez@usc.edu (G.A.L.); djgamble@usc.edu (D.J.G.); 2Department of Orthopaedic Surgery, Keck School of Medicine, University of Southern California, 1540 Alcazar, Los Angeles, CA 90089, USA; hudnall@usc.edu

**Keywords:** regeneration, lizard, blastema, hedgehog signaling, chondrogenesis, organoid

## Abstract

(1) Background: Lizard tail regeneration provides a unique model of blastema-based tissue regeneration for large-scale appendage replacement in amniotes. Green anole lizard (*Anolis carolinensis)* blastemas contain fibroblastic connective tissue cells (FCTCs), which respond to hedgehog signaling to create cartilage in vivo. However, an in vitro model of the blastema has not previously been achieved in culture. (2) Methods: By testing two adapted tissue dissociation protocols and two optimized media formulations, lizard tail FCTCs were pelleted in vitro and grown in a micromass blastema organoid culture. Pellets were analyzed by histology and in situ hybridization for FCTC and cartilage markers alongside staged original and regenerating lizard tails. (3) Results: Using an optimized serum-free media and a trypsin- and collagenase II-based dissociation protocol, micromass blastema organoids were formed. Organoid cultures expressed FCTC marker CDH11 and produced cartilage in response to hedgehog signaling in vitro, mimicking in vivo blastema and tail regeneration. (4) Conclusions: Lizard tail blastema regeneration can be modeled in vitro using micromass organoid culture, recapitulating in vivo FCTC marker expression patterns and chondrogenic potential.

## 1. Introduction

Lizards are the closest evolutionary relatives to mammals with the ability to perform large-scale appendage regeneration [1,2]. As amniotes, lizards share many developmental milestones with mammals, delineating them from traditional amphibian models of limb and tail regeneration, such as the salamander [1,3,4]. Green anole lizards, *Anolis carolinensis*, share this capacity to regenerate tails naturally through epimorphic or blastema-based regeneration [2,5,6]. Interestingly, lizards regenerate an “imperfect” copy of their tails, producing an unsegmented cartilaginous tube rather than a patterned, ossified vertebra, providing a valuable model for large-scale cartilage regeneration, an ability humans notably lack [1,2,7].

Upon amputation, anoles regenerate their tails over the course of 28 days, forming immune-privileged blastemas, heterogenous collections of connective tissue and muscle progenitor cells in various states of differentiation by day 14 (D14) [1,5,8,9]. Sonic hedgehog signaling (Shh) produced by invading regenerating spinal cords activate a cartilage program in surrounding blastema cells. Undifferentiated blastema cells begin to express Sox9 and differentiate into chondrocytes. As regeneration continues, collagen type 2 alpha chain 1+ (Col2a1^+^) cartilage tubes form surrounding spinal cords as tails elongate. Meanwhile, other blastema cells differentiate into muscle, fat, blood vessel, dermis, and other key tissue types in regenerated tails [2,5,10,11,12]. Treatment with exogenous Shh agonist (SAG) in vivo results in ectopic cartilage formation in blastemas, demonstrating the capacity for most, if not all, blastema cells to take on a cartilage program [7].

While lizard tail cartilage has been well studied in a number of species [6,11,13,14,15] the specific cell populations that give rise to blastema cells with chondrogenic potential have yet to be isolated. In the past, intervertebral disc, periosteum and other connective tissues have all been studied as potential sources of regenerated lizard tail cartilage [6,7,10,11,13]. Furthermore, clues from salamander studies may aid in identification of blastema cell populations responsible for cartilage formation in regenerate lizard tails. Axolotl limb blastema cells express paired related homeobox 1, PRRX1, a pan-fibroblastic connective tissue cell (FCTC) marker and exhibit molecular funneling towards a common dedifferentiated state during blastema formation. Over the course of regeneration, these cells then re-differentiate into cartilage, skeleton, periskeletal cells, and regenerated fibroblastic connective tissues [16,17]. We hypothesize that similar cell types and biological processes regulate lizard tail blastema formation and differentiation. To follow lizard tissue tail differentiation, we have developed an in vitro system where growth conditions can be controlled and monitored.

Micromass and organoid cultures allow for in vitro modeling of in vivo cellular processes. Micromass culture, which involves high density cell seeding and aggregation, has been effectively utilized with various species and cell types to explore chondrogenic potential and in vitro models of cartilage development [18,19,20,21,22,23]. Here, we have developed a novel lizard blastema organoid micromass culture system through the optimization of enzymatic digestion and culture conditions. Lizard blastema stem cell isolation and growth were tested by using two combinations of enzymatic digestion buffers and two culture media formulations previously reported as pro-chondrogenic. Lizard blastema organoids were tested for their expression of FCTC markers and for their chondrogenic potential in response to hedgehog stimulation. 

## 2. Materials and Methods

### 2.1. Lizard Maintenance and Handling

Wild adult green anole lizards (*Anolis carolinensis*, Carolina Biological Supply Company, Burlington, NC, USA; LLL Reptile, Chandler, AZ, USA) were maintained 25.5 °C with 65% humidity on a 12 h light: 12 h dark schedule with 12 h of 50 W basking heat lamp and UVB lamp (Zoo Med, San Luis, Obispo, CA, USA) treatment during light hours. Care and experimental use of animals was conducted in accordance with USC Institutional Animal Care and Use Committee (IACUC) approved protocol #20992.

### 2.2. SAG Injections

Lizards were injected with 20 µg Smoothened Agonist HCl (SAG) (Selleck Chem, Houston, TX, USA) per gram anole body weight every other day, beginning on the first day of amputation day and continuing until tail collection.

### 2.3. Lizard Tail Amputations and Dissociation Protocols

Lizard tails were amputated to begin regeneration and collected in Hank’s balanced salt solution (HBSS) (ThermoFisher, Waltham, MA, USA) supplemented with 100 units/mL penicillin and 100 µg/mL streptomycin (Sigma, St. Louis, MO, USA) (HBSS with P/S) at day 0 (D0), day 14 (D14) or day 28 (D28) time points for histology and cell dissociation. Original D0 tails were triple washed with Betadine (Henry Schein, Melville, NY, USA) followed by triple rinses of tap water and HBSS with P/S. Tails were incubated for 45 min with agitation in HBSS with P/S and 0.1% ethylenediaminetetraacetic acid (EDTA) (ThermoFisher, Waltham, MA, USA). The epidermis was removed with forceps and discarded. The remaining tissues were minced in HBSS with P/S. 

Two dissociation protocols were optimized. The first, herein the Trypsin protocol, was adapted from the Mello and Tuan 1999 embryonic chick limb bud cell isolation protocol [20]. Trypsin protocol dissociation solution was formulated with 1 mg/mL collagenase II (Worthington Biochemicals, Lakewood, NJ, USA) and 1 mg/mL Trypsin (Gibco ThermoFisher, Waltham, MA, USA) in HBSS with P/S and filtered with a 0.22 µM Steriflip filter (MilliporeSigma, Burlington, MA, USA). Minced tail pieces were added to solution and incubated at 37 °C for 45 min with agitation and manual pipetting every 15 min. Dissociation was stopped with FBS (Gibco ThermoFisher, Waltham, MA, USA).

The second dissociation protocol, herein the Dispase protocol, was adapted from Farmer et al., 2021 [24] coronal suture dissociations. Dispase protocol dissociation solution was formulated with 3 mg/mL collagenase II and 4 units/mL Dispase (Corning, Corning, NY, USA) in HBSS with P/S. Minced tail pieces were added to the solution and incubated at 37 °C for 45 min with agitation and manual pipetting every 15 min. Dissociation was stopped with 30% FBS and 6 mM CaCl2 (Sigma, St. Louis, MO, USA) in 1× Phosphate-buffered saline (PBS) (Gibco ThermoFisher, Waltham, MA, USA).

Following dissociation in both protocols, cells were filtered with 40 µM basket filters (Corning, Corning, NY, USA) and plated in 96 well V-bottom plates (Corning, Corning, NY, USA) with 1 million cells per well. Cells were pelleted via centrifugation at 500× *g* for 10 min.

### 2.4. Cell Culture

Cell pellets were maintained in either mammalian media or avian media for 5 weeks. Mammalian media [25] were formulated with 0.1 µM dexamethasone (Sigma, St. Louis, MO, USA), 40 µg/mL proline (Sigma, St. Louis, MO, USA), 10 µg/mL ITS+ (Life Technologies ThermoFisher, Waltham, MA, USA), 50 µg/mL ascorbic acid (Sigma, St. Louis, MO, USA), 100 units/mL penicillin, 100 µg/mL streptomycin, and 250 ng/mL fungizone antimycotic (Life Technologies ThermoFisher, Waltham, MA, USA) in Dulbecco’s Modified Eagle Media (DMEM)/Ham’s F12 with 1× Glutamax (Gibco ThermoFisher, Waltham, MA, USA). Avian media [20] were formulated with 10% FBS, 1% glucose (Sigma, St. Louis, MO, USA), 1.1 mM CaCl2, 2.5 mM beta-glycerophosphate (Sigma, St. Louis, MO, USA), 50 µg/mL ascorbic acid (Sigma, St. Louis, MO, USA), 100 units/mL penicillin and 100 µg/ mL streptomycin (Life Technologies ThermoFisher, Waltham, MA, USA) in DMEM/Ham’s F12 with 1× Glutamax. Both mammalian and avian media conditions were supplemented with 1 nM SAG. Media were changed every other day. 

### 2.5. Histology

Lizard tails were collected, fixed overnight in 4% paraformaldehyde (PFA) (Electron Microscopy Sciences, Hatfield, PA, USA) and then decalcified for 1 week in Osteosoft (Sigma, St. Louis, MO, USA). Tails underwent a sucrose (Sigma, St. Louis, MO, USA) gradient and were frozen in Optimal Cutting Temperature Compound (OCT) (Fisher Scientific, Hampton, NH, USA). Tail cryoblocks were sectioned at 16 µM thickness. 

Cell pellets were fixed in V-bottom plates for 30 min in 4% PFA. Pellets underwent a sucrose gradient and were frozen in OCT. Pellet cryoblocks were sectioned at 10 µM thickness.

### 2.6. In Situ Hybridization (ISH)

ISH was performed using the RNAscope 2.5 HD Detection Kit (RED) and proprietary ISH probes (Advanced Cell Diagnostics, Newark, CA, USA) [26]. Samples were baked for 1 h at 60 °C, rinsed in 1× PBS and post-fixed in 4% PFA at 4 °C for 15 min. Slides were dehydrated in an ethanol (VWR, Visalia, CA, USA) gradient and allowed to dry for 5 min. Samples were incubated in hydrogen peroxide for 10 min and then allowed to dry before outlining with PAP pen (Vector Laboratories, Burlingame, CA, USA). Slides were incubated in protease solution at 40 °C for 30 min. Then, slides were hybridized with ISH CDH11(fibroblastic connective tissue), Sox9 (cartilage program), Col2a1 (cartilage) or negative control bacteria DapB probes at 40 °C for 2 h. The probe signal was amplified with 4 proprietary amplification reagents at 40 °C for 1.5 h and another 2 amplification reagents at room temperature for 45 min. The signal was detected with FAST RED solution (1:60 FAST RED B:FAST RED A solution) for 10 min at room temperature, revealing puncta in red for analysis. Slides were counterstained with 50% Gill’s I hematoxylin (StatLab, McKinney, TX, USA) and 0.02% Ammonium Hydroxide (Sigma, St. Louis, MO, USA). Slides were mounted in xylene (VWR, Visalia, CA, USA) and EcoMount (Biocare, Pacheco, CA, USA). Slides were imaged on a Keyence BX800 microscope (Keyence, Itasca, IL, USA) in brightfield.

## 3. Results

### 3.1. Fibroblastic Connective Tissue Cells Express CDH11 in Lizard Blastema

We sought to identify fibroblastic connective tissue cells (FCTCs) in lizard tail blastemas, similar to PRRX1^+^ FCTCs in salamander limb [16]. Connective tissues in original tails (D0) exhibited CDH11 expression via histology/RNAscope in situ hybridization in epidermis and periosteum (Figure 1A–A’’) compared to bacterial gene DapB probe negative control (Appendix A). During D14 blastema stages, CDH11 expression specifically marked blastema cells (Figure 1B,B’), while differentiated muscle bundles within blastemas exhibited markedly lower CDH11 expression (Figure 1B’’) Low residual CDH11 expression within regenerating muscle is expected due to mesenchymal muscle stem cell signature within the regenerating bundles [27]. Upon tail regrowth (D28), CDH11^+^ FCTCs again localized to the epidermis and perichondrium of regenerated tails (Figure 1C–C’’). Taken together, these results indicate that CDH11^+^ cells exist in original tail connective tissues and form the majority of blastema tissue during tail regrowth. After regeneration is completed, CDH11 expression is again restricted to differentiated connective tissues, mimicking original tail expression patterns.

### 3.2. CDH11^+^ Blastema Cell Culture System Optimization

Next, we aimed to develop culture conditions for in vitro micromass blastema organoids. Given the abundance of CDH11^+^ cells detected in periosteum described above, we adapted a protocol from embryonic mouse coronal suture cell dissociations [24] involving 4 units/mL Dispase and 3 mg/mL collagenase II, herein the Dispase protocol. Additionally, we adapted a protocol from chick limb bud cell dissociations, often comparable to the lizard tail bud in developmental studies [20], containing 1 mg/mL trypsin and 1 mg/mL collagenase II, herein the Trypsin protocol. 

Original lizard tails were isolated and dissociated by utilizing either Dispase or Trypsin protocols and pelleted via centrifugation in V-bottom plates. Culture media for traditional mammalian serum-free chondrogenic media were tested [25], herein mammalian media, containing 0.1 µM dexamethasone, 40 µg/mL proline, 10 µg/mL ITS+, 50 µg/mL ascorbic acid, 100 units/mL penicillin, 100 µg/mL streptomycin and 250 ng/mL fungizone antimycotic in DMEM/Ham’s F12 with 1 mM Glutamax. Additionally, culture media for chick limb bud micromass culture were tested [20], herein avian media, containing 10% FBS, 1% glucose, 1.1 mM CaCl_2_, 2.5 mM β-glycerophosphate, 50 µg/mL ascorbic acid, 100 units/mL penicillin and 100 µg/mL streptomycin in DMEM/Ham’s F12 with 1 mM Glutamax. Both culture media were supplemented with 1 nM smoothened agonist (SAG) to mimic Shh signals received from regenerating spinal cords in blastemas during in vivo lizard tail regeneration. 

After 5 weeks in culture to mimic full tail regeneration, pellets were fixed, sectioned, and analyzed via RNAscope in situ hybridization and histology. Cells isolated via the Dispase protocol did not form one solid pellet in mammalian media culture, instead forming several smaller pellets that lined plate wells (Figure 2A). All other conditions yielded single pellets (Figure 2B–D). Cells isolated via the Dispase protocol and cultured in mammalian media also did not show uniform expression of CDH11 (Figure 2A,A’) compared to bacterial gene DapB probe negative control (Appendix A), in contrast to the other conditions that yielded pellets with more uniform CDH11 expression of cells not obscured by pigmented cells (Figure 2B–D’). Cells isolated via the Trypsin protocol and cultured in mammalian media (Figure 2C,C’) showed the highest and most uniform CDH11^+^ cells in culture and, therefore, recapitulated in vivo blastema most accurately in terms CDH11 expression patterns.

### 3.3. Sox9 Expression in Wild-Type and SAG-Treated Blastema

During natural lizard tail regeneration, regenerated spinal cords invade blastemas, supplying Shh signals to blastema cells. Hedgehog stimulation primes surrounding blastema cells to activate a Sox9^+^ chondrogenic program, resulting in blastema cell differentiation into chondrocytes. Original tail tissues (Figure 3A,A’) and blastema cells lateral to regenerating spinal cords (Figure 3A’’’) do not receive and respond to endogenous Shh signaling due to their distance from the spinal cord. In contrast, blastema cells medial to the regenerating spinal cord (Figure 3A’’) do receive Shh signals and express high levels of Sox9, signaling the activation of chondrogenic programming.

Lizards were systemically treated with hedgehog agonist SAG (Figure 3B). When treated with exogenous SAG in vivo, original tail FCTCs remained unaffected by Shh signaling and did not show Sox9^+^ activation (Figure 3B’). SAG treatment causes blastema cells both lateral (Figure 3B’’) and medial (Figure 3B’’’) to regenerating spinal cords to express high levels of Sox9. Thus, exogenous Shh signaling can activate Sox9^+^ chondrogenic programming in blastema FCTCs regardless of location, while original FCTCs remain unaffected by additional signals. These results indicate fundamental differences in chondrogenic potential between CDH11^+^ blastema cells and original tail FCTCs.

### 3.4. Micromass Blastema Organoid Cultures Mimic Regenerating Tail Cartilage Formation 

Lizards treated with SAG-regenerate tails exhibit ectopic cartilage regions (Figure 4A) made up of Col2a1^+^ chondrocytes (Figure 4A’). In culture, cells isolated via Dispase and Trypsin protocols recapitulated this phenomenon when cultured in mammalian media, forming Col2a1^+^ cartilage in vitro (Figure 4B,C). The same cells exhibited low Col2a1 expression in avian media and did not appear to form fully differentiated cartilage (Figure 4D,E).

Combined with observations of pellet morphology and CDH11 expression (Figure 2), these results suggest that the Trypsin protocol combined with mammalian media produced the most accurate recapitulation of regenerate lizard tail cartilage formation in vitro. Cells isolated via the Dispase protocol cultured in mammalian media did not form a single pellet in culture and, thus, do not mimic blastema cell masses in vitro, while Dispase cells cultured in avian media did not display uniform CDH11 expression. Cells isolated via the Trypsin protocol and cultured in mammalian media formed single cell pellets in vitro with uniform CDH11 expression, modeling the D14 blastema, and formed fully differentiated Col2a1^+^ cartilage over the same time period of regeneration as in vivo lizard tails. 

## 4. Discussion

This study demonstrates the chondrogenic potential of lizard tail FCTCs using a novel organoid model of lizard tail blastema development. Organoid models have emerged as important techniques for reducing complicated biological processes to their most vital cell populations for study in vitro [23]. In doing so, organoid models facilitate the interrogation of simulated tissue homeostasis or pathologies with drug and genetic treatments that would be impossible in vivo. However, new organoid models must first be validated to ensure they faithfully recreate known biological process before they can be confidently used to interrogate new phenomena. Here, we confirmed our micromass lizard blastema organoids form cartilage in response to the same signals regulating regenerated tail chondrogenesis in vivo. We have previously defined the lizard blastema cell state as one that undergoes chondrogenesis in response to hedgehog signaling [7,10,28,29]. Here, we validated that organoids formed from tail FCTC populations formed new cartilage in response to hedgehog signaling.

Prior to this study, the cellular identities contributing to chondrogenic blastema cells were unknown. Here, we identified CDH11 as an effective marker for lizard FCTC populations that contribute to tail blastemas, and this study adds to the growing body of literature supporting FCTCs as a (the) main contributor of appendage blastemas [16,17,30]. For example, PRRX1^+^ salamander limb FCTCs contribute to blastemas during salamander limb regrowth [16]. Cre-based lineage tracing experiments suggested that PRRX1^+^ FCTCs from multiple mesodermal tissues de-differentiated into a common blastema cell state before re-differentiating into new limb tissues, including cartilage. The lizard PRRX1 gene remains poorly annotated, but CDH11 was identified as an acceptable substitute for identifying FCTC populations. The histology/ISH results presented indicate that nearly all non-muscle blastema cells highly express CDH11, including those that condense to form the regenerated tail cartilaginous skeleton. Taken together, these results suggested that chondrogenic blastema cells are derived from CDH11^+^ resident FCTC populations. 

Ideally, transgenic tissue-specific reporter lines would be used to trace the differentiation fates of CDH11^+^ lizard FCTC cells though blastema formation and tail regeneration. However, the realities of reptile reproduction, including late-developmental stage oviposition, make transgenic lizard generation much more difficult than salamander genetic engineering, and the feasibility of lizard gene reporter line establishment remains prohibitively difficult for basic research [31]. To overcome these challenges, we employed selection by different enzymatic digestions and culture system to select for CDH11^+^ cells to demonstrate their chondrogenic capabilities in vitro. Specifically, we found that digestion by trypsin and collagenase II enzymes and culture under serum-free conditions selected for CDH11^+^ FCTCs. These cells underwent chondrogenesis in response to hedgehog stimulation, fulfilling our definition of lizard blastema cells. However, further work is needed to determine the exact mechanisms by which hedgehog stimulations result in cartilage formation. For example, does hedgehog stimulation result in increased cartilage formation through differentiation of uncommitted FCTCs or proliferation of specific FCTC populations pre-biased towards chondrogenesis? Additionally, further work is needed to confirm if CDH11^+^ FCTCs are the only cell populations responding to hedgehog signaling during regeneration, given the known response of mesenchymal stem cells and lizard satellite cells to activate chondrogenic programming in response to exogenous Shh [32,33]. 

This study also uncovered potential differences between lizard tail and salamander limb blastema cell derivation. As previously mentioned, recent studies with salamander limb regeneration show a cellular funneling of FCTCs to a more stem-like state during the salamander limb blastema derivation process before differentiating into cartilage [16]. However, our results suggest that lizard FCTCs do not have to proceed through the blastema cell process to form cartilage. Instead, FCTCs isolated directly from original tails form cartilage in response to hedgehog stimulation, bypassing the blastema cell stage. However, our results do suggest differences in chondrogenic potential between FCTCs within original tail tissues and blastemas. We show that, when lizards are systemically stimulated with hedgehog agonist SAG, only blastema FCTCs undergo chondrogenesis. No cartilage formation is detected in FCTC populations within original tail portions. Thus, instead of the reprogramming that takes place during salamander limb blastema formation, we now hypothesize that mechanisms exist within original tail tissues that suppress hedgehog-induced chondrogenesis in resident lizard FCTC populations. Given the importance of hedgehog signaling in mammalian skeletal and limb development [34], repressive molecular mechanisms may keep FCTCs in an adult state, where liberating these cells result in a developmental signature that allows for response to hedgehog signals, resulting in the activation chondrogenic programming. During blastema formation in vivo or micromass organoid culture in vitro, FCTCs are freed from their niches and their inhibitors and allowed to condense and form cartilage following hedgehog stimulation. Future studies will investigate these novel topics by studying and comparing the epigenetic states of FCTCs within original tails and blastemas. Furthermore, we will study the role of cell–cell contacts in FCTC condensation and chondrogenesis in vivo and in vitro, allowing for better understanding of cartilaginous and skeletal development in all organisms with developmental hedgehog signaling. 

In summary, this manuscript has established an organoid model of regenerated lizard tail blastema formation and chondrogenesis. The dependency of the model on different selection method through different enzymatic digestions are presented, revealing different isolated cell populations. Similarly, the effects on culture media conditions were tested. In the end, only isolation and culture conditions that resulted in a high number of CDH11^+^ FCTCs resulted in organoid models capable of undergoing chondrogenesis in response to hedgehog stimulation. These results point to the FCTC origin of lizard blastema cells and has laid the foundation for future assessments of the effect of starting cell populations on cartilage phenotype outcomes. For example, we have previously known that there are two distinct zones of regenerated lizard tail cartilage, each with distinct cell sources and developmental trajectories. Proximal lizard tail cartilage is directly derived from periosteal cells and undergoes hypertrophy and endochondral ossification, similarly to mammalian cartilage fracture calluses. Distal regenerated lizard tail cartilage forms from blastema cells and resist hypertrophy and ossification. Both proximal and distal cartilage regions form in response to hedgehog signaling [7]. Since we have shown here that periosteal cells, blastema cells, and organoid cells are CDH11-positive, it is currently unclear whether the cartilage formed in organoid model is representative of distal or proximal cartilage or both. Future work will be aimed at co-staining and testing under hypertrophy conditions [35] to see if they can undergo cartilage hypertrophy and terminal differentiation. 

## Figures and Tables

**Figure 1 jdb-10-00012-f001:**
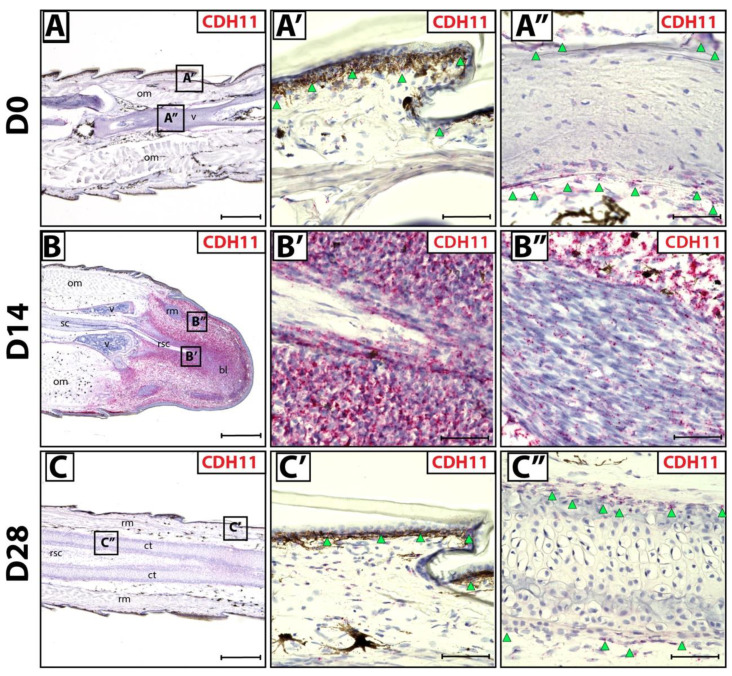
CDH11^+^ cells in original and regenerating lizard tails. (**A**) Original tail (D0), (**B**) blastema (D14) and (**C**) regenerated tail (D28) analyzed by histology/in situ hybridization for CDH11 expression. (**A’**,**A’’**) Higher magnification views of original tail regions identified in Panel **A** highlighting CDH11 expression in (**A’**) epidermis (green arrowheads) and (**A’’**) periosteum (green arrowheads). (**B’**,**B’’**) Higher magnification views of tail blastema regions identified in Panel **B** contrasting (**B’**) high CDH11 expression in blastema connective tissue and (**B’’**) low expression in blastema muscle bundles. (**C’**,**C’’**) Higher magnification views of regenerated tail regions identified in Panel **C** including (**C’**) regenerated epidermis (green arrowheads) and (**C’’**) perichondrium (green arrowheads). (**A**–**C**) Scale bar = 500 µM. (**A’**–**C’’**) Scale bar = 50 µM. bl—blastema; ct—cartilage tube; om—original muscle; rm—regenerated muscle; rsc—regenerated spinal cord; sc—spinal cord; v—vertebrae.

**Figure 2 jdb-10-00012-f002:**
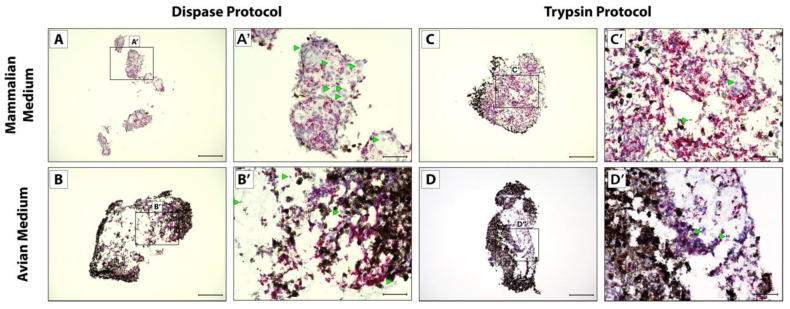
CDH11 expression in micromass blastema organoid cultures. Cell pellets derived from Dispase dissociation protocol (**A**–**B’**) or Trypsin Dissociation protocol (**C**–**D’**) and cultured in (**A**,**A’**,**C**,**C’**) mammalian media or (**B**,**B’**,**D**,**D’**) avian media were analyzed by histology/in situ hybridization for CDH11 expression. (**A’**,**B’**,**C’**,**D’**) Higher magnification of pellet in Panels (**A**), (**B**), (**C**), and (**D**), respectively. Cells with notable lack of CDH11 are highlighted (green arrowheads). (**A**,**B**,**C**,**D**) Scale bar = 200 µM. (**A’**,**B’**,**C’**,**D’**) Scale bar = 50 µM.

**Figure 3 jdb-10-00012-f003:**
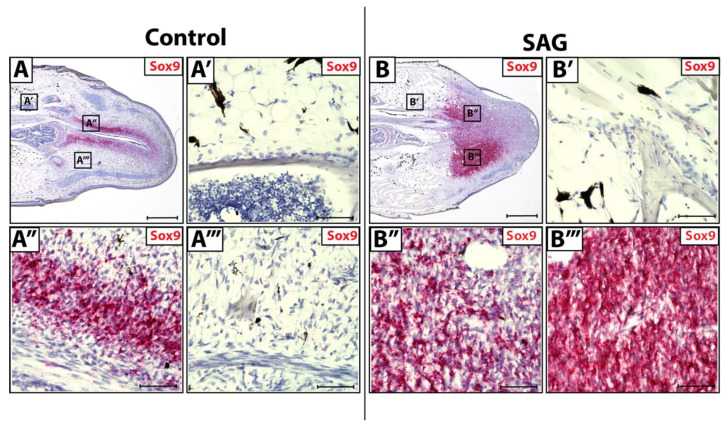
Sox9 expression in control and SAG-treated blastema tails. (**A**) Control and (**B**) SAG-treated blastemas (D14) analyzed by histology/in situ hybridization for Sox9 expression (**A’**–**A’’’**) Higher magnification of regions identified in Panel **A** highlighting (**A’**) original tail FCTCs, (**A’’**) blastema FCTCs medial to regenerated spinal cord and (**A’’’**) blastema FCTCs lateral to the regenerating spinal cord. (**B’**–**B’’’**) Higher magnification of regions marked in Panel **B** showing (**B’**) original tail FCTCs, (**B’’**) medial and (**B’’’**) lateral blastema FCTCs with respect to the regenerating spinal cord. (**A**,**B**) Scale bar = 500 µM. (**A’**–**A’’’**,**B’**–**B’’’**) Scale bar = 50 µM.

**Figure 4 jdb-10-00012-f004:**
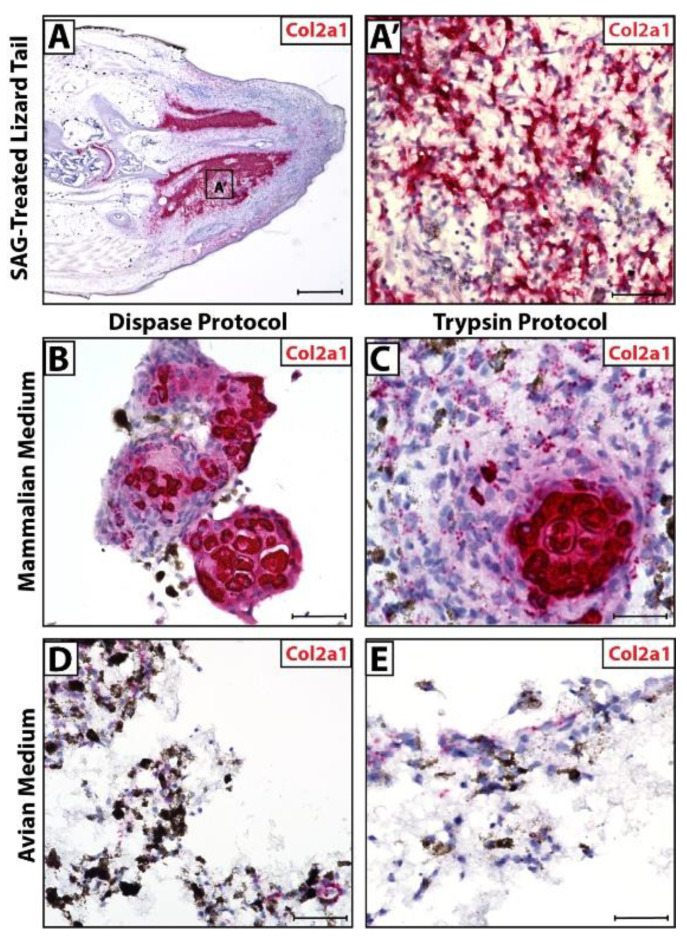
Micromass blastema organoid cultures mimic lizard tail regenerate cartilage formation in optimized dissociation protocol and media. (**A**) SAG-treated regenerated lizard tail (D28) analyzed via histology/in situ hybridization for Col2a1 expression. (**B**) Higher magnification of region identified in Panel **A** highlighting Col2a1^+^ ectopic cartilage region. (**B**–**E**) Col2a1 expression in cell pellets dissociated using (**B**,**D**) Dispase protocol or (**C**,**E**) Trypsin protocol, cultured in (**B**,**C**) mammalian media or (**D**,**E**) avian media and analyzed via histology/in situ hybridization for Col2a1 expression. (**A**) Scale bar = 500 µM. (**A’**,**B**–**E**) Scale bar = 50 µM.

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
