# Peer review of "Lizard Blastema Organoid Model Recapitulates Regenerated Tail Chondrogenesis"

_jdb, 2022, doi:10.3390/jdb10010012_

Round 1
Reviewer 1 Report
Major Concerns
- The results and outcomes presented here, require further validation through the gene and protein expression data. The Authors are requested to incorporate those results to substantiate the data procured.
- The representative figures provided for all the results need to be revisited in terms of the quality of the images (high resolution required, currently the images are pixelating).
- In the present manuscript, figures are uniformly lacking clear labels, scale bars/image magnification data, and internal controls for histology. This data must be added diligently.
- The significance of using two different methods for dissociation of cells (Trypsin and Dispase methods) followed by the derived conclusions are missing in this manuscript
Specific comments
- Line 68: PRRX1 appears for the first time. Therefore, instead of the abbreviation, the complete word should be used.
- Line 176: Grey arrowheads are mentioned in the figure legend, but are not clearly visible in the image. Also, the colors for displaying the target molecules (e.g., CDH11) in red font in the image are merged with the histological output image. Changing the font colors here is highly recommended.
- Segregating ICC and IHC results will provide more clarity and help in the correlation of the data.
- Line 222: Demarcation between figures 3 A’ and B’ are missing (Figure 3)
- Line 222-228: Blastemal cells labeled with Sox9 are mentioned to be either medial or lateral blastemal cells. Nevertheless, the rationales for such conclusions are missing. Adding references might help.
- Line 244: The font colors for figure numbers (A, B, C, D, E, and F) or the molecule targeted are not clear (Figure 4).
- Line 253-254: The rationale for selecting Trypsin protocol over the Dispase protocol is unclear.
- Line 288 to 291: “However, the realities of reptile reproduction, including late-developmental stage oviposition, make transgenic lizard generation much more difficult that salamander genetic engineering, and the feasibility of lizard reporter line establishment has not yet been established by current techniques [26]” revisit statement for correctness.
- Line 306-307: “Instead, FCTCs isolated directly from original, uninjured tails form cartilage in response to hedgehog stimulation.” revisit statement, it seems unclear.
- Also, the correlation between the experiments conducted, their outcomes, and derived conclusions is lacking.
Reviewer 2 Report
Ms by Vonk et al. is focused on the role of FCTC positive stem cells in the regenerating tail A carolinensis lizards and isolated cells in an “organoid” culture in vitro. These are exciting data, designed to examine the presence of FCTC+ cells in the blastema and their contribution to chondrogenesis.
Specific comments:
- Figure 1, is missing an important control, there are no scrambled or nonsense probes used in ISH. This should be done. Also in B’’, there appear to be single cells that look CDH11 positive in the muscle. The authors should comment on this.
- The images in figure 2 are poor quality however, they do not support the author’s statement in lines 200-205 regarding the uniformity of CDH11 expression.
- In figure 3 there is a lack of scrambled or nonspecific control.
- The use of SAG induces ectopic expression of Sox9 (compare A to B). SAG known to induce the expression of chondrogenesis pathway genes in adipose derived MSCs, making it possible that more than one cell type is responding to this drug within the regenerating tail. Without some double label methodology it is too narrow a conclusion that only the FCTC cells are participating.
- Can the ectopically activated cells in SAG treated tails all respond by activating the chondrogenic program? Furthermore, related to specificity of cell type, how limited are the FCTC and ectopically activated cells in terms of cell fate.
- It would be more convincing if other genes in the pathway were also detected such as the Gli proteins and expression of the ECM proteins associated with chondrogenesis.
- Isolated Pax7+ cells from the lizard can express the cartilage program in micromass culture, that highly resembles the “organoid” culture described in this manuscript. What makes this procedure an organoid and how does it differ from the previously published work?
